# Mapping of Monodominant *Gilbertiodendron dewevrei* Forest Across the Western Congo Basin Using Sentinel-2 Imagery

Ellen Heimpel [1,2,*], David J. Harris [1], Josérald Mamboueni [3], David Morgan [4], Crickette Sanz [5,6] and Antje Ahrends [1]

1    Royal Botanic Garden Edinburgh, Edinburgh EH3 5LR, UK; dharris@rbge.org.uk (D.J.H.); aahrends@rbge.org.uk (A.A.)
2    School of Geosciences, University of Edinburgh, Edinburgh EH8 8XP, UK
3    Institut National de Recherche Forestière, Brazzaville BP 10787, Congo; manbuenijo4@gmail.com
4    Lester E. Fisher Center for the Study and Conservation of Apes, Lincoln Park Zoo, Chicago, IL 60614, USA; dmorgan@lpzoo.org
5    Department of Anthropology, Washington University in St. Louis, St. Louis, MO 63130, USA; csanz@wustl.edu
6    Congo Program, Wildlife Conservation Society, Brazzaville BP 14537, Congo
*    Correspondence: eheimpel@rbge.org.uk or ellen.heimpel@ed.ac.uk

**Abstract:** Tropical rainforests are complex mosaics of different forests types, each with its own biodiversity and structure. Efforts to characterize and map diversity and composition of tropical forests are vital at both local and larger scales in order to improve conservation strategies and accurately monitor anthropogenic threats. However, despite advances in remote sensing, classifying and mapping forest types remains a significant challenge and remotely sensed classifications in the tropics often treat forests as a single category. Here, we used Sentinel-2 data, and a high-quality ground reference dataset, to map monodominant *Gilbertiodendron dewevrei* forest, a unique forest type in central Africa. We used a random forest classifier, and spectral, vegetation, and textural indices, to map *G. dewevrei* forest across the Sangha Trinational, a network of national parks in central Africa. The overall accuracy of our classification was 83% when evaluated against an independently sampled reference test dataset, successfully distinguishing this monodominant forest from the spectrally similar *terre firme* mixed forest present throughout much of the study area. The gray level co-occurrence matrix (GLCM) textural metrics proved the most important factors for distinguishing *G. dewevrei* forest, due to the homogenous canopy texture created by this monodominant species. In conclusion, our study illustrates that freely available Sentinel-2 data hold promise for mapping distinct forest types in tropical forests, particularly when they exhibit structural and textural differences, as seen in monodominant and mixed forests, and provided that high-quality ground reference data are available.

**Keywords:** Sentinel-2; Google Earth Engine; random forest; *Gilbertiodendron dewevrei*; monodominance; Congo Basin

## 1. Introduction

Tropical rainforests form complex mosaics of distinct vegetation types, which can vary substantially in species composition, structure, and function across small spatial scales [1–4]. Despite advances in remote sensing [5–9], accurately classifying and mapping these forest types remains a significant challenge, with distributions of forest types still poorly known in tropical forests, e.g., [10–12]. Consequently, remote sensing efforts still often classify tropical forests as a single category, e.g., [13], limiting our ability to quantify the distribution

and extent of different rainforest types, to identify the underlying environmental drivers, and to design specific management and conservation strategies [1,14]. The absence of maps with a higher-class resolution of forest types also constrains efforts to monitor anthropogenic threats such as selective logging. Changes in species assemblages and ecosystem function due to human activities [15] may therefore go undetected, hampering targeted conservation interventions.

This is of particular concern in central Africa, where there are forests of global conservation priority, due to their high biodiversity [16–19], extensive intact areas of forest [19,20], large mammal populations [21–23], carbon stocks [24,25], provision of income for nations [22], and livelihoods for local communities [26–28]. These forests are increasingly threatened by industrial activities and related socioeconomic developments, including agriculture [29], forestry [22,30], and mining [31]; as well as increasing climatic threats of decreased rainfall and higher temperatures [32,33]. A lack of on-the-ground data in central Africa hampers the accurate monitoring of these globally important forests.

The main challenges that currently constrain efforts to classify tropical rainforests from satellite images are insufficient on-the-ground data [34], the difficulty of establishing characteristic spectral fingerprints for individual species in such highly diverse systems [35], and the low availability of cloud-free images across the tropics [36,37]. In addition, in tropical rainforests, and possibly particularly central Africa, a further challenge is presented by striping in images, caused by the stitching together of adjacent swaths of satellite imagery with different viewing angles [38].

A few key studies have highlighted that increasingly, spectral and textural information contained within satellite data has the potential to differentiate different forest types within the dense mosaic of the Congo Basin forests [39,40]. In particular, there is an increasing opportunity with the availability of Sentinel-2 data, which have fine spatial resolution (10–60 m), global coverage, rich spectral information, and short revisit periods [41]. The dense time series of Sentinel-2 imagery that has built up is also decreasing the barrier of persistent cloud cover. A few studies have attempted the use of Sentinel-2 imagery to map forest types across central Africa, including Dalimier et al. [42], who mapped 13 forest types on a large scale using both pixel-based and object-based classification; and Picard et al. [12], who mapped six forest types in a study area in the northern Republic of Congo using deep learning architectures.

Here, we focus on a forest system in central Africa, *Gilbertiodendron dewevrei* monodominant forest. This is a type of tropical lowland rainforest dominated by the leguminous tree *Gilbertiodendron dewevrei*. This species dominates in the canopy, making up 60–90% of the canopy level trees, and creating its own ecosystem through environmental filtering [43–50]. Heimpel et al. [51] show that this forest type harbors a unique diversity of vascular plant species, and differs structurally from other forest types. They highlight the need for its separate consideration within conservation policy measures and carbon modelling. However, monodominant *G. dewevrei* forest has low spectral separability from adjacent mixed *terre firme* forest. For example, Barbier et al. [52] could successfully automate the mapping of *G. dewevrei* forest with extremely high-resolution data, but not with spot6–7 images with a spatial resolution of just over 2 m. Previous studies have had some success mapping monodominant forest using multi-spectral Landsat imagery. Degagne et al. [53] mapped the distribution of monodominant *Dicymbe corymbosa* forest in Guyana's upper Portaro River Basin, achieving user accuracies of above 80%, and Helmer et al. [54] found that a decision tree-based classification of gap-filled Landsat images could distinguish *Mora excelsa* forests from other forest types in Trinidad and Tobago, due to the smoother and taller canopy created by the dominance of this species.

This study investigates whether Sentinel-2 multi-spectral imagery can be used to produce a high-resolution map of monodominant *Gilbertiodendron dewevrei* forest, a forest type in central Africa that shows low spectral separability from the surrounding mix of spectrally very similar forest types. We used a high-quality ground reference dataset, supplemented by points collected from high-resolution satellite imagery. We dealt with striping issues within Sentinel-2 imagery by including multiple spectral, vegetation, and textural inputs to the classification, removing those most influenced by the striping; and by incorporating large amounts of reference data. The study produced a map of *G. dewevrei* forest across the study site of the Sangha Trinational, a network of national parks in the Republic of Congo, Cameroon, and the Central African Republic, and investigated the importance of different predictors for separating *G. dewevrei* from surrounding forest types, including individual spectral bands, vegetation indices, and textural metrics, to guide future studies attempting to separate monodominant tropical forest types.

## 2. Materials and Methods

The workflow for this study involved the use of a random forest algorithm in Google Earth Engine, trained on 1378 reference points, to classify a Sentinel-2 image composite for monodominant *Gilbertiodendron dewevrei* forest. The overall workflow is summarized in Figure 1.

### 2.1. Study Site

This study was conducted in the Sangha Trinational, a network of protected areas in the north-west of the Congo River basin, across Cameroon, the Central African Republic, and the Republic of Congo (Figure 2). It covers a total area of 746,309 hectares, including three national parks: the Nouabalé-Ndoki National Park (Republic of Congo; 2°05′–3°03′N, 16°51′–16°56′E, 4238.7 km$^2$), the Lobéké National Park (Cameroon; 2°05′–2°30′N, 15°33′–16°11′E, 2178.54 km$^2$), and the Dzanga-Ndoki National Park (Central African Republic; 2°22′–3°08′N, 16°06′–16°55′E, 1143.26 km$^2$), which is split across two distinct units. There is also a buffer zone around the national parks that includes the Dzanga-Sanga Forest Reserve. For this study, we focus only on the three national parks.

Annual rainfall within the Sangha Trinational ranges from 1450 to more than 1600 mm [49,55]. Soils within the region are classified as Ferralsols (both Xanthic and Orthic) and Orthic Luvisols [56]. White [16] classified the vegetation of this area broadly as mixed moist semi-evergreen Guineo-Congolian rainforest, and Harris [57] identified five distinct forest types: mixed species *terre firme* forest, monodominant *G. dewevrei* forest, *Raphia* swamp forest, streamside forest, and seasonally flooded forest.

### 2.2. Dataset

#### 2.2.1. Reference Data Points

In order to produce an accurate map of *G. dewevrei* forest, we collected a high-quality dataset of reference points from the Sangha Trinational and surrounding areas to train the classifier. Reference data consisted of points in *G. dewevrei* forest and points collected in other areas of tree cover—primarily mixed *terre firme* forest, as well as secondary regrowth, seasonally flooded forests, blackwater flooded forests, oil palm plantations, coffee plantations, other agriculture, swamps, *Raphia* swamps, etc. This data consisted of a total of 497 points in *G. dewevrei* forest, and 881 points in other areas of tree cover. These were primarily field observations, supplemented by data points from herbarium specimens and those derived from high-resolution satellite imagery.

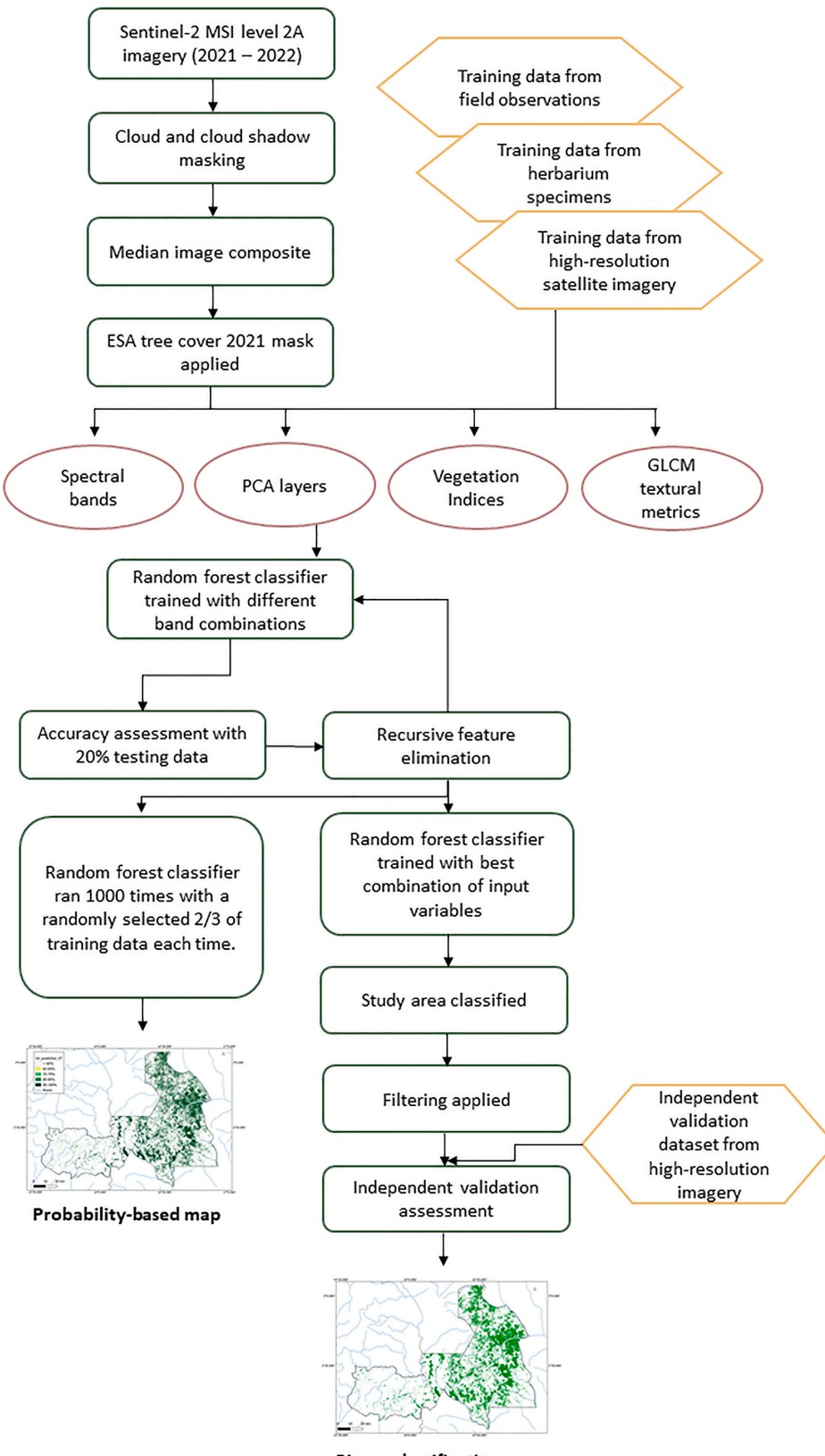

**Figure 1.** Workflow diagram summarizing the datasets and methods used to produce maps of monodominant *Gilbertiodendron dewevrei* forest across the Sangha Trinational.

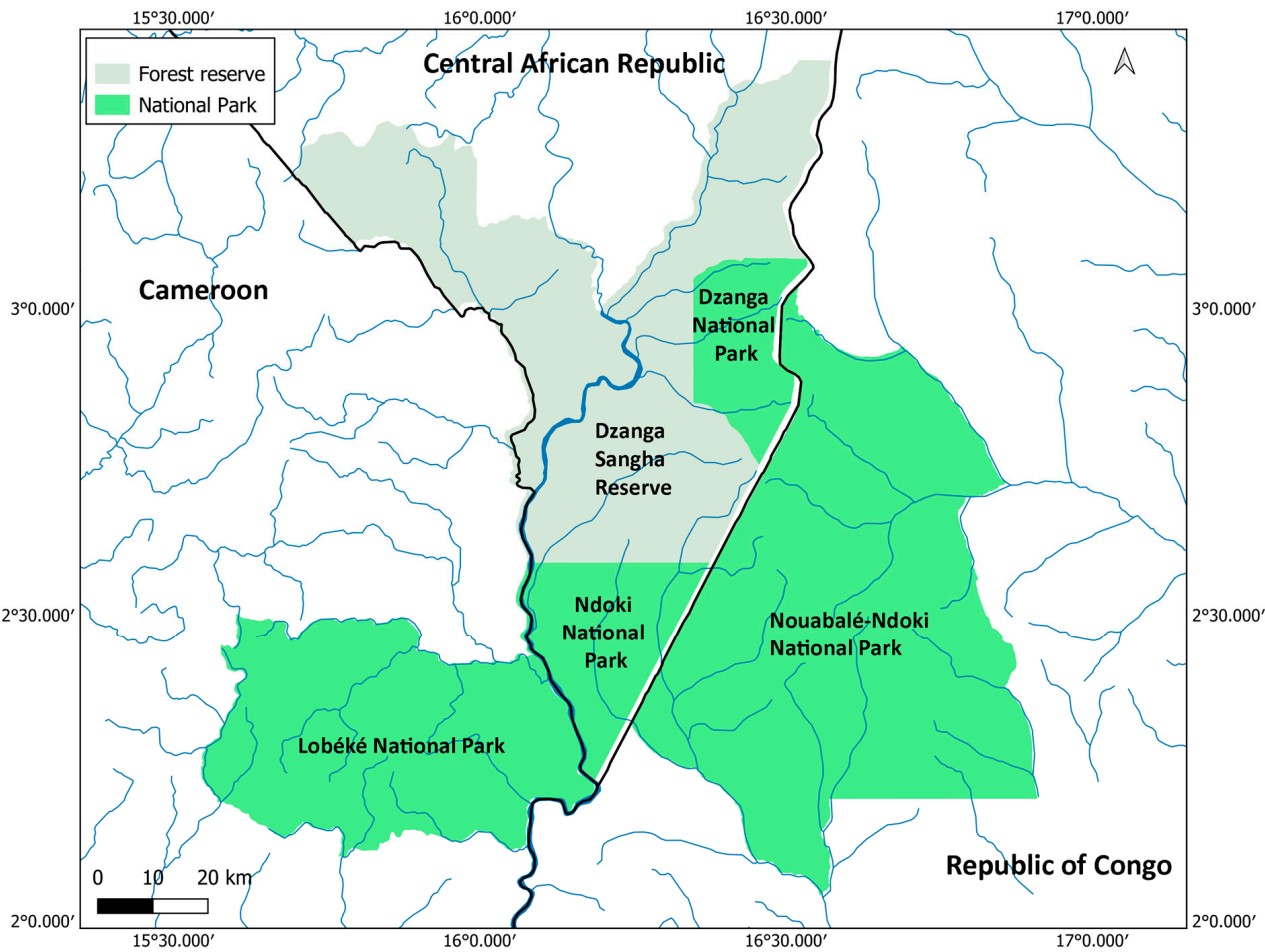

**Figure 2.** Map of the Sangha Trinational. Blue lines represent major rivers, and black lines represent country borders. Created using QGIS v3.34.

Field observations were collected on the ground, using hand-held GPS units (Garmin GPSMap 65s), and consisted of 291 points in *G. dewevrei* forest and 110 in mixed *terre firme* forest. These field data were supplemented by plotting coordinates from geo-referenced herbarium specimens collected in *G. dewevrei* forest, and in other forest types, which were visually examined and verified using high-resolution satellite imagery. Visual interpretation was carried out using a custom-built project in Collect Earth Online [58], which provided access to a very high-resolution Mapbox satellite imagery base map. We also used high-resolution imagery from Google Earth Pro and Microsoft Bing Maps. Only points for which we had strong confidence were selected. Additional points in both *G. dewevrei* forest and other areas of tree cover were selected from high-resolution satellite imagery alone. Example points are shown in Figure 3, and a map of the training points is presented in Appendix A (Figure A1).

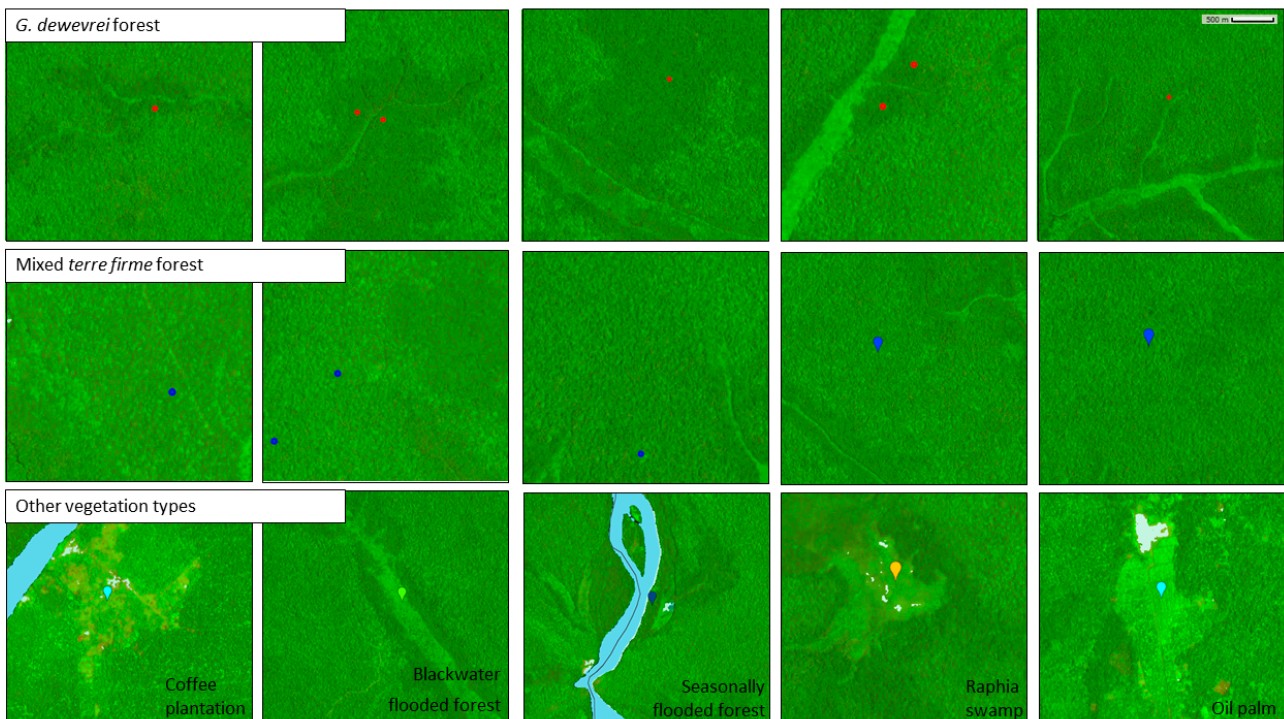

**Figure 3.** Sentinel-2 image composite in Google Earth Engine showing example reference points. Top panel shows points in *G. dewevrei* forest (red), middle panel shows points in mixed *terre firme* forest (blue), and bottom panel shows examples of other categories of tree cover included in the classification. Background satellite imagery is an image composite from Sentinel-2 images for 2021–2022, visualized using B11 (SWIR1), B8 (NIR), and B4 (red band).

### 2.2.2. Sentinel-2 Images for Classification

Sentinel-2 imagery was downloaded from the study area for the period of 2021–2022. We used level-2A images, which have undergone atmospheric correction using Sen2Cor. Multi-year composites were used to reduce the effects of cloud cover, and artifacts caused by atmospheric correction models in cloudy regions. A phenology approach was also tested, but this did not improve our classifier and hence was not pursued. To remove clouds and cloud shadows, we used the QA60 cloud mask band and Sentinel-2 cloud probability datasets. Pixels were selected with a cloud probability of less than 10%. We then created an image composite using the median value. To restrict classification to areas with tree cover, we applied a tree cover mask from the ESA WorldCover 10 m 2020 product, which provides a global land cover map for 2020 at 10 m resolution [59]. According to an independent evaluation, the ESA global landcover map achieves good accuracies for tree cover (user accuracy of $80.1 \pm 0.1$ 95% CI and producer accuracy of $89.9 \pm 0.1$ 95% CI) [60].

### 2.3. Calculating Spectral, Vegetation and Textural Indices

We computed the following candidate predictors for the classification: spectral bands, spectral indices, gray level co-occurrence matrix (GLCM) textural indices and a PCA for each band of Sentinel-2 imagery. GLCM textural metrics represent the distance and angular relationships between sub-regions of an image, by quantifying the frequency at which, in a mobile kernel of a specified size, a pair of grayscale pixel brightness values occurs [61]. These were calculated in Google Earth Engine using ee.Image.glcmTexture with a neighborhood size of 7, and kernel size of a $3 \times 3$ square. Principle component analysis was also carried out in Google Earth Engine, in order to capture variability in spectral signatures, following methods proposed by Gutkin et al. [62]. This involved centring the data, computing the covariance matrix, extracting eigenvalues and eigenvectors, and transforming the bands

into principal components, which were then normalized by their standard deviation. The final inputs to classification were selected using recursive feature elimination [63], whereby the classification was run iteratively, removing the least important input variable at each step. Training and validation accuracies were calculated using a withheld 20% subset of the training data and plotted (Figure A2) to evaluate the performance of the classification. A total of 16 input bands and indices were chosen as the fewest number of variables that still achieved a validation accuracy of 0.98 (Table 1). These included GLCM sum average (SAVG) texture, the standard deviation of NDVI, and PCAs of Sentinel bands.

**Table 1.** Input variables to random forest classification of Sentinel-2 imagery for monodominant *G. dewevrei* forest in the Sangha Trinational.

| Input | Description |
|---|---|
| B3 | Sentinel-2 Green band (10 m) |
| B5 | Sentinel-2 Red Edge 1 band (20 m) |
| B6 | Sentinel-2 Red Edge 2 band (20 m) |
| B8A | Sentinel-2 Red Edge 4 band (20 m) |
| B9 | Sentinel-2 Water vapour band (60 m) |
| B11 | Sentinel-2 SWIR 1 band (20 m) |
| B12 | Sentinel-2 SWIR 2 band (20 m) |
| B3_PCA | PCA of B3 (Green band) |
| B5_PCA | PCA of B5 (Red Edge 1 band) |
| NDVI | Normalized difference vegetation index: $NIR = (B8 - B4)/(B8 + B4)$ |
| SATVI | Soil-adjusted total vegetation index: $SATVI = ((SWIR1 - RED)/(SWIR1 + RED + 0.1)) \times (1.1 - (SWIR2/2))$ |
| texture | Standard deviation of NDVI ($5 \times 5$ pixel moving window) |
| savg B11 | Sum average B11. Sum average = average of pixel pairs within a GLCM. $Savg = \sum_{i=2}^{2Ng} iP_{x+y}(i)$ Where Ng is the number of distinct gray levels in the quantized image. |
| savg B5 | Sum average B5 |
| savg B6 | Sum average B6 |
| savg B8 | Sum average B8 |

### 2.4. Random Forest Classifications

In order to produce a binary classification map of *G. dewevrei* forest, we ran a random forest algorithm with 500 decision trees on our Sentinel-2 image composite, using the reference data as input. Random forest was selected as the algorithm is generally robust to multi-collinearity and can effectively handle a large number of predictor variables. Random forest was chosen over deep learning methods due the computational intensity of deep learning, and the black box nature of these techniques leading to a loss of interpretability [64]. Support vector machine (SVM) was considered alongside random forest, as both these methods are frequently used in the remote sensing community; however, random forest was selected since it allows the study of variable importance [65], does not come with the added complexity of choosing a suitable kernel [66], and in a review of remote sensing literature was reported as achieving higher accuracy and lower variance for studies of LULC [64].

A sensitivity analysis of the random forest algorithm was carried out, testing the number of trees iteratively from 100 to 1000, and calculating training and validation accuracies using a withheld 20% of the training data (Table A2). We selected 500 trees for the final classification, as maximum training accuracy was achieved, and validation accuracy had effectively stabilized. This is a common parameter selection in random forest

classifications [67]. The resulting classification was filtered to remove noise, removing areas with fewer than 16 connected pixels. Further filtering was carried out on the finished map by running majority filters twice with kernels of size $7 \times 7$.

To produce a probability-based classification, we implemented stratified two-thirds Monte Carlo selection, whereby 1000 times, two-thirds of all data points per class were randomly selected and the random forest classification was run. Each pixel was then plotted based on the proportion of times it was identified as *G. dewevrei* forest. This was loosely based on methods set out by Crezee et al. [68] when mapping peat thickness in the Congo basin.

### 2.5. Accuracy Assessments

We ran accuracy assessments on the binary classification map, following recommended good practices [69]. Given that splitting data into training and validation data carries a risk of lack of independence, and may lead to overestimating accuracy, we created a new independent validation dataset, by randomly selecting 300 points across our study area, 100 that had been classified as *G. dewevrei* forest and 200 that had been classified as other areas of tree cover. These points were then visually classified independently using high-resolution satellite imagery in Collect Earth Online. We compared these validation points to the classified map following methods proposed by Olofsson et al. [69], to obtain an error matrix, an error matrix of estimated areas, and user, producer, and overall accuracy figures. A map of the validation points is presented in Appendix A (Figure A1).

### 2.6. Linking the Map with Environmental Data

To investigate the association of *G. dewevrei* forest with watercourses, and to demonstrate the usability of this map, we mapped rivers across the study area using WWF hydroSHEDS Free flowing Rivers Networks v1 [70]. A buffer of 250 m was created around every river, and the proportion of mapped *G. dewevrei* forest that fell within this area was calculated.

## 3. Results

### 3.1. Classification Maps

We produced a binary classification of monodominant *Gilbertiodendron dewevrei* forest versus other forest types across the National Parks of the Sangha Trinational, using a random forest classification algorithm on a 2021–2022 Sentinel-2 mosaic (Figure 4). This algorithm classified the study area as 2287 km$^2$ *G. dewevrei* forest and 5571 km$^2$ other forest area; thus, the mapped area suggests that forest area within the Sangha Trinational is 29.1% monodominant *Gilbertiodendron dewevrei* forest.

In addition to the binary map, we also produced a probability-based classification by running the random forest classification 1000 times, using a different subset of the training data each time and then rendering the map based on the proportion of times each pixel was identified as *G. dewevrei* forest (Figure 5). This map showed a similar general pattern of *G. dewevrei* distribution, but it picked up on smaller patches with a lower probability of being dominated by *G. dewevrei*, and highlighted that the boundary pixels have a lower probability. Shapefiles for the binary map and probability-based map are provided (Supplementary Materials).

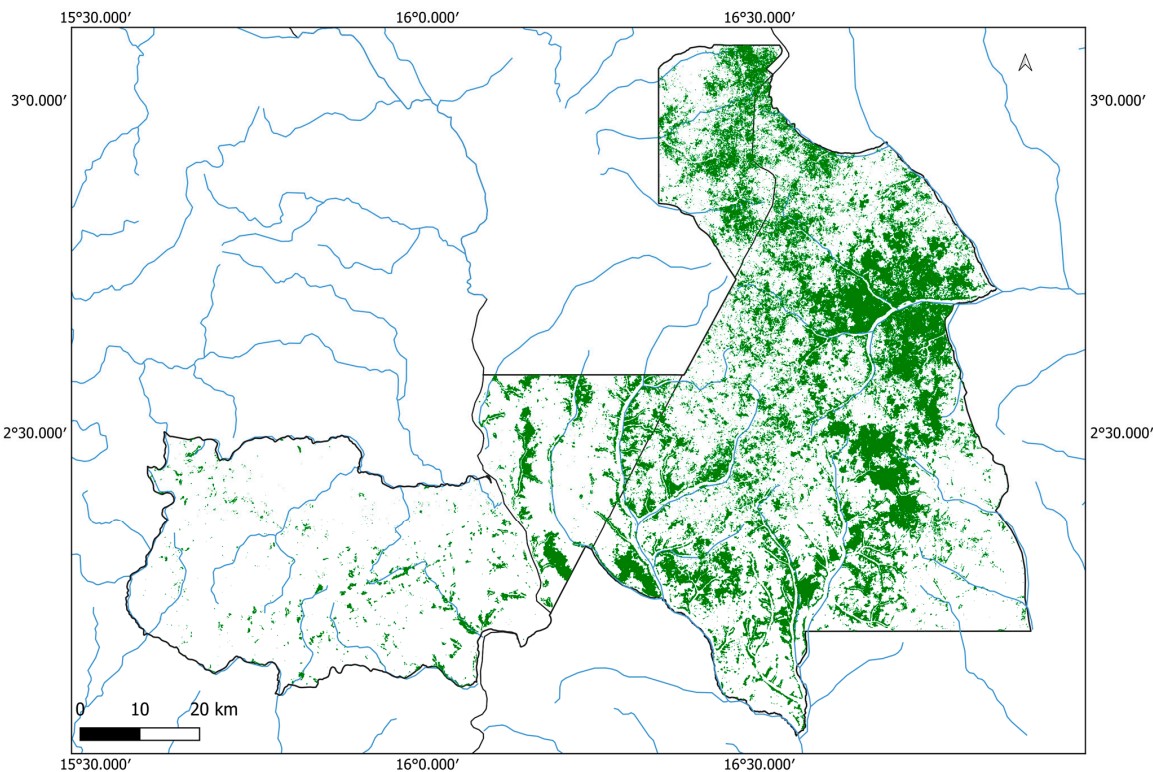

**Figure 4.** Binary classification of *Gilbertiodendron dewevrei* monodominant forest distribution across the Sangha Trinational. Dark green patches represent *G. dewevrei* forest, black outlines show national park and country boundaries, and blue represents large rivers in the area. Created using QGIS v.3.34.

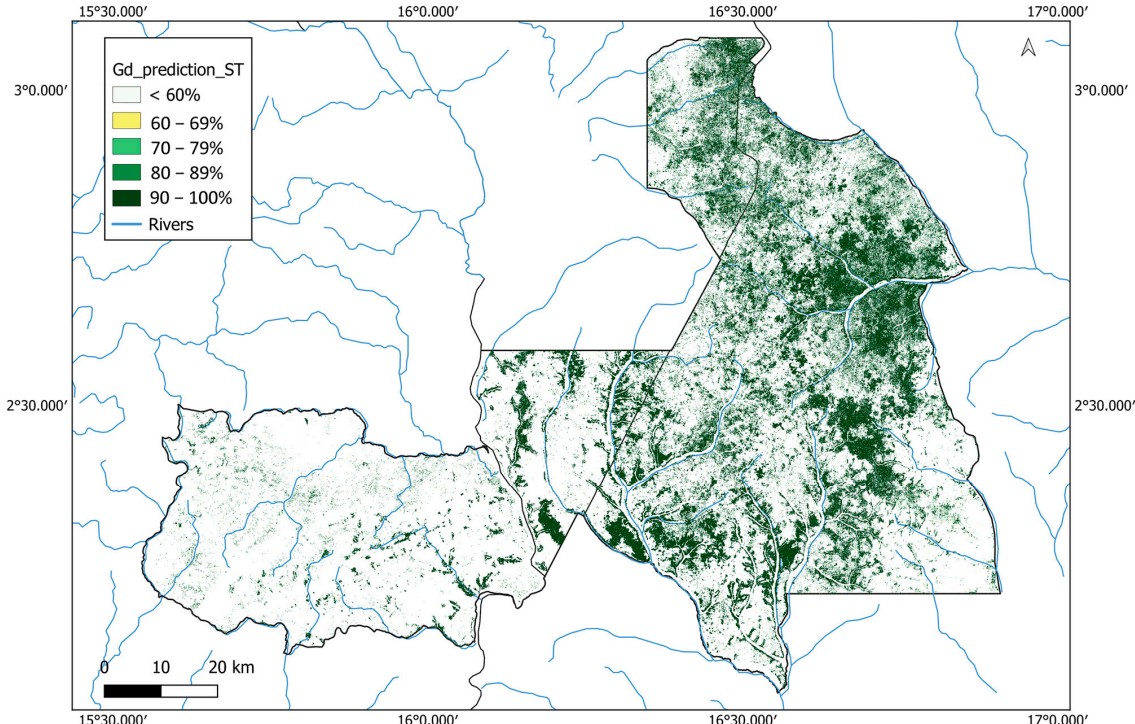

**Figure 5.** Probability-based classification of monodominant *Gilbertiodendron dewevrei* forest across the Sangha Trinational. *Gilbertiodendron dewevrei* forest pixels are plotted according to the percentage of times they were classified as *G. dewevrei* in a classification run 1000 times with different combinations of training data. Black outlines show national park and country boundaries, and blue represents large rivers in the area. Created using QGIS v.3.34.

### 3.2. Classification Accuracy

Overall accuracy was 0.83 [Cl = 0.04], with user accuracy for *G. dewevrei* classification of 0.80 [CI = 0.09], and producer accuracy of 0.67 [Cl = 0.07] (adjusted by area weights, following Olofsson et al. [69]) (Table 2). The mapped area of *G. dewevrei* forest was lower than the estimated area (228,700 ha compared to 272,559 ha), indicating that our mapped area of *G. dewevrei* forest is conservative. The actual proportional area covered by *G. dewevrei* forest may be 34.7% [CI 30.4–39.0%].

**Table 2.** Accuracy assessment and area estimation for mapping of *Gilbertiodendron dewevrei* forest across the Sangha Trinational according to methods proposed by Olofsson et al. [69].

| | (A) Error Matrix of Reference Data | | | | | |
|---|---|---|---|---|---|---|
| | | | **Reference** | | | |
| | | *G. dewevrei* | Other | **Total** | **Area** | **Precision** |
| **Map** | *G. dewevrei* | 65 | 16 | 81 | 228,700 | 0.80 |
| | Other | 35 | 184 | 219 | 557,100 | 0.84 |
| | **Total** | 100 | 200 | 300 | 785,800 | |
| | **Recall** | 0.65 | 0.92 | | | |
| | **F1 score** | 0.72 | 0.88 | | | |

| | (B) Error Matrix of Area Proportions | | | | | |
|---|---|---|---|---|---|---|
| | | | **Reference** | | | |
| | | *G. dewevrei* | Other | **Total** | **User accuracy** | **95% CI** |
| **Map** | *G. dewevrei* | 0.234 | 0.057 | 0.291 | 0.8 | 0.09 |
| | Other | 0.113 | 0.596 | 0.709 | 0.84 | 0.05 |
| | **Total** | 0.347 | 0.653 | 1 | | |
| | **Producer accuracy** | 0.67 | 0.91 | **Overall** | **0.83** | **0.04** |
| | **95% CI** | 0.07 | 0.04 | | | |

| | (C) Mapped and Estimated Area | | | |
|---|---|---|---|---|
| | **Mapped area (ha)** | **Estimated area (ha)** | **Lower 95% CI** | **Upper 95% CI** |
| *G. dewevrei* | 228,700 | 272,559 | 238,907 | 306,211 |
| Other | 557,100 | 513,241 | 479,589 | 546,893 |
| **Total** | 785,800 | | | |

To establish the sensitivity of the accuracy figures to uncertainty and errors in the class labelling, we also present the accuracy values that we obtained if we assumed that every point in our validation dataset of which we were not 100% confident was in fact the alternate class (Table A1). This resulted in negligible changes to our user and producer accuracy figures, and no change to overall accuracy.

### 3.3. Striping in Satellite Imagery

We used level 2A Bottom-Of-Atmosphere (BOA) Sentinel-2 imagery, which has undergone atmospheric correction. This should also ensure a bidirectional reflectance distribution function (BRDF)-corrected product. However, the L2A products currently available from ESA do not fully resolve the issue in the difference in brightness across the acquisitions in the west–east direction [38]. This is an artifact caused by the double acquisition of sentinel images [71]. The striping effect caused by this is apparent in our satellite image composite (Figure 6A), and calls into question the accuracy of the classification for this section (Figure 6B). However, accuracy assessments within the stripe report similar accuracies to that of the whole map, and that of the non-striped region (Tables A3 and A4), demonstrating we have minimized the potential residual influence of the striping on our classification.

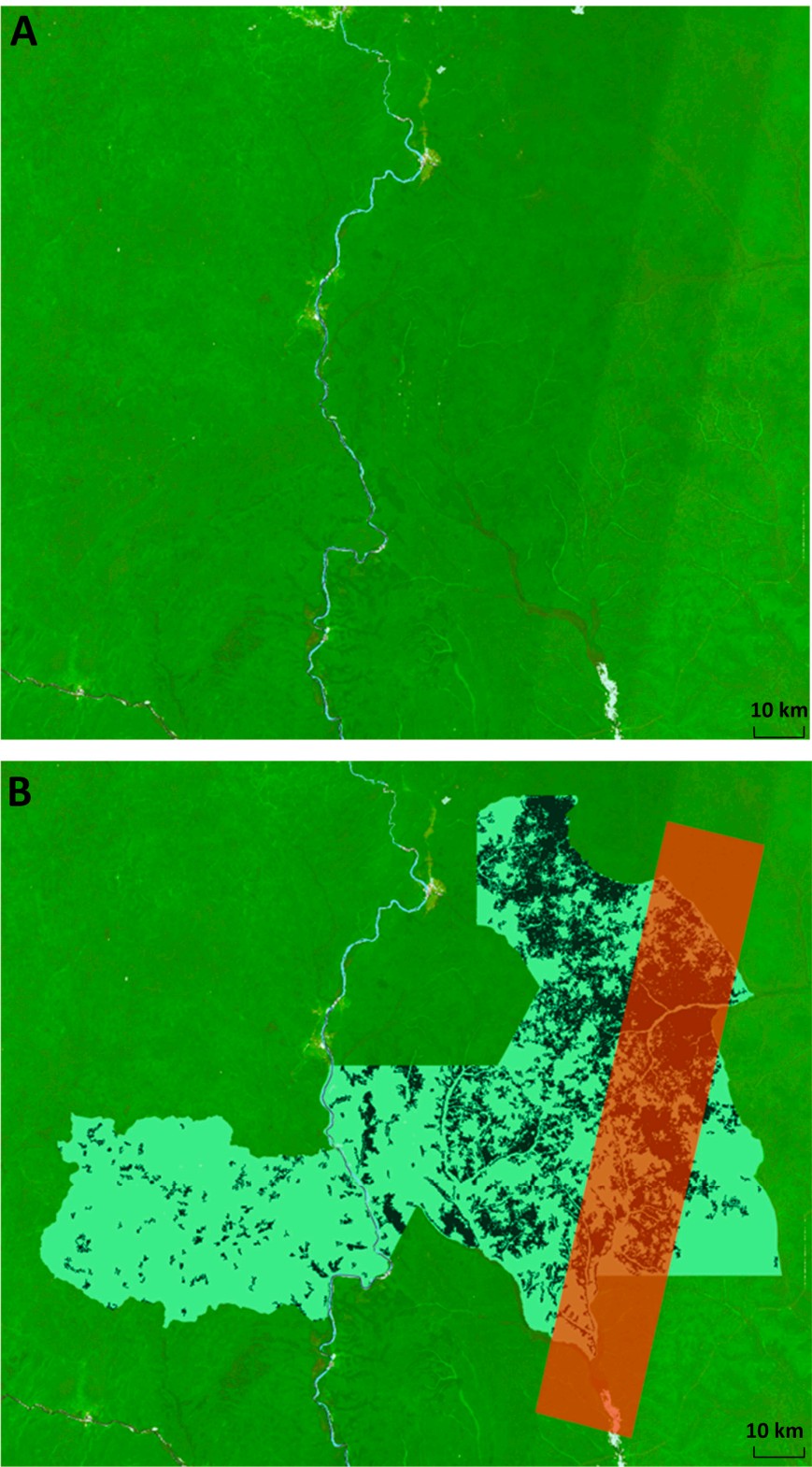

**Figure 6.** Striping in Sentinel-2 imagery caused by adjacent swaths from the two Sentinel satellites at different viewing angles. (**A**) Sentinel- 2 image composite for 2021–2022 showing striped portion. Visualized in Google Earth Engine using B11 (SWIR1), B8 (NIR) and B4 (red band). Non-forest pixels are masked out according to the ESA global landcover map. (**B**) Classified map of the Sangha Trinational in Google Earth Engine, with the region of the classification affected by striping in the Sentinel-2 composite highlighted with a red polygon.

### 3.4. Variable Importance

We performed recursive feature elimination to remove redundant layers, leaving only those input variables contributing most to classification accuracy. Textural metrics made the greatest contribution to classification accuracy scores, in particular savgB6 and savgB11 (Figure 7).

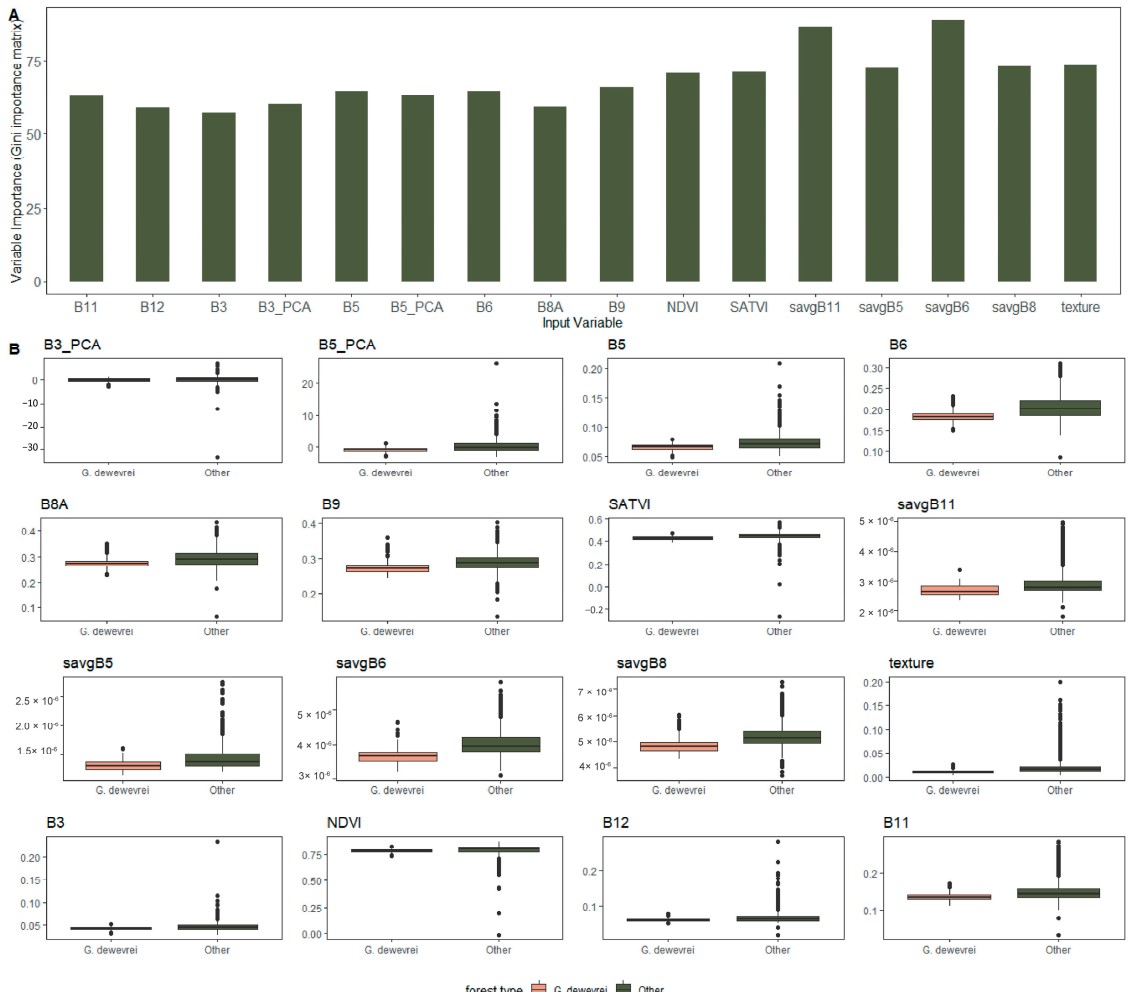

**Figure 7.** (**A**) Relative importance (Gini importance matrix), and (**B**) differences between values for *G. dewevrei* forest and other forest types, for input variables of random forest classification of Sentinel-2 imagery across the Sangha Trinational for monodominant *Gilbertiodendron dewevrei* forest.

### 3.5. Linking Gilbertiodendron dewevrei Distribution to Environmental Variables

Our binary classification map (Figure 4) confirms that *G. dewevrei* forest is often found along the borders of rivers and watercourses. However, there are also large areas of connected pixels that extend beyond rivers. We conducted an analysis to investigate how much of our mapped *G. dewevrei* forest was found in close proximity to a river, finding that only 11.4% of *G. dewevrei* forest in the Sangha Trinational was within 250 m of a river (Figure 8). The dataset of watercourses used here is a global one, whose resolution does not go down to the smallest streams. Therefore, it is likely this figure is a slight underestimation.

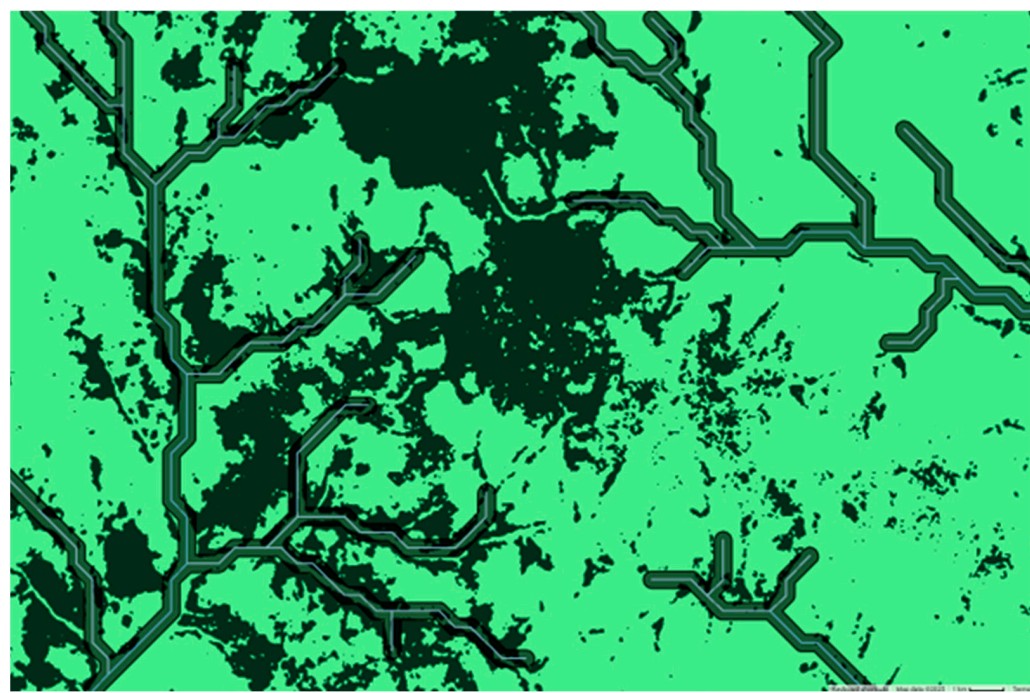

**Figure 8.** Section of river buffer analysis conducted in Google Earth Engine. A global dataset of watercourses was plotted (WWF hydroSHEDS Free Flowing Rivers Networks v1), and a 250 m buffer was drawn around each river. The proportion of *G. dewevrei* forest that fell within these buffers was then calculated. The dark green color shows *G. dewevrei* forest and the light green is other forest types.

## 4. Discussion

Here, we map monodominant *Gilbertiodendron dewevrei* forest across the Sangha Trinational based on Sentinel-2 imagery, and 1378 reference points, with an overall accuracy of 0.83. The most important inputs were two textural metrics: sum average of B11, and sum average of B6, suggesting that the uniform texture created by the many crowns of similar shape and width is key to the mapping of monodominant *G. dewevrei* forest. Sum average (SAVG) is the average of pixel pairs within a GLCM [72], and this metric has also been identified by other studies as useful for forest applications such as mapping forest disturbance [73], mapping oil palm [74], and for detecting selective logging [75].

Remote sensing imagery has previously shown promise for mapping monodominant forests. Letouzey [76] began mapping *G. dewevrei* forest in Cameroon using visual interpretation of aerial photographs. Degagne et al. [53] found Landsat imagery was able to delineate monodominant *D. corymbosa* forest in Guyana with an accuracy of more than 80%. Barbier et al. [52] used a set of high spatial resolution satellite images (including spot6 and 7 images, Pleiades, Orbview, Quickbird, and GeoEye images) to manually map areas dominated by *G. dewevrei.* More recently, Picard [12] mapped out the vegetation types in an area of 18,500 km$^2$ to the south of our study area in the Republic of Congo, of which a small proportion was *G. dewevrei* forest, using sentinel-2 imagery and deep learning architectures. Here, we build on this work by focusing on *G. dewevrei* forest in an area with much greater distribution of this forest type (29% compared to 2%), and investigating more spectral and textural inputs to the classification.

The current study maps 29% of the Sangha Trinational as monodominant *G. dewevrei* forest. This is significantly higher than estimates in the literature for the Sangha Trinational (e.g., 11% in [77]), and for other areas across the Congo Basin [12,76,78]. We also map higher quantities of *G. dewevrei* forest in the Nouabalé-Ndoki national park in northern Republic of Congo, and the Dzanga-Ndoki national park in the Central African Republic,

than in the Lobéké National Park in eastern Cameroon. This fits with previous mapping and observations from Letouzey [76]. Extending this classification over a larger area would provide further insights into the underlying drivers of current day distribution of *G. dewevrei*, including soils, climate, and past human activity.

*Gilbertiodendron dewevrei* forest is reported as occurring mainly along riversides [1,48]. Kearsley et al. [48] argue, in their study area west of Kisangani, DR Congo, that this is due to the low drought resistance and low potential for water regulation of *G. dewevrei* trees. However, other studies have noted areas of *G. dewevrei* forest on hilltops away from water courses [49,52,79,80]. Here, we find that only 11.4% of the *G. dewevrei* forest in the Sangha Trinational is within 250 m of a river. This contrasts with Kearsley's et al. [48] theory, and also findings by Picard et al. [12], who investigated a proxy for water table depth, HAND (height above nearest drainage) index, which suggested that *G. dewevrei* forest is distributed where the water table is shallow. However, *G. dewevrei* forest is much more represented in our study area, making up almost 30% of forest cover in the Sangha Trinational, which may explain the differences in distribution patterns. *G. dewevrei* trees have a large taproot [50], which can reach a depth of approximately 2 m, potentially helping it access deep groundwater and grow in these higher elevation areas [49]. Further work could investigate the differences between *G. dewevrei* forest in the upland patches and streamside *G. dewevrei* forest, both through looking for differences in spectral signatures, and fieldwork investigating ecological factors such as soil.

Mapping of specific forest types, as presented here, has the potential to inform local conservation management and planning. *G. dewevrei* forest in the Sangha Trinational is distinct from mixed species forest [51], and has apparent old-growth status [81]. At present, indications are that, if cleared, this forest type may take thousands of years to grow back, due to its slow growth rate, extremely low dispersal rate, and the importance of feedback mechanisms for the maintenance of its monodominance [43,44,46,47,49]. This map will inform management of *G. dewevrei* forest within the national parks, and can be extended to the surrounding areas in which logging, hunting, and harvesting of some non-timber forest products is permitted.

Our study was impacted by striping in Sentinel-2 imagery, an artifact caused by the double acquisition of Sentinel satellites. This effect is especially pronounced over dense humid forests [38]. In initial iterations, the striping effect was influencing our classification, leading to an over-classification of *G. dewevrei* forest within the striped area. We addressed this by removing certain inputs from the classification that were most affected by the striping, achieving comparable accuracies to that in the non-striped region. However, when the classification was extended to areas where less training data were available (e.g., 150 km$^2$ in all directions from our study area), the over-classification of these pixels to the east and west of the adjacent swaths as *G. dewevrei* forest was still apparent. We therefore advise caution in expanding classifiers over larger regions where high-quality training data are not available. Other authors have cited this as a problem, especially across tropical central Africa, e.g., [38,42], suggesting various fixes, which although they do improve the quality of the imagery, do not completely remove the effect. The need for complex corrections for the geometry of the sensor undermines efforts to make remote sensing products accessible, and ESA needs to provide a product for which this effect is addressed and accounted for.

In this study, we present both a binary classification, and a probability-based classification. These were similar; however, there were more small patches of *G. dewevrei* forest in the probability-based classification. This was partly due to the lack of filtering for this map, as pixels did not exist in discreet classes. Most smaller patches of *G. dewevrei* had a lower probability (often 60–70%), suggesting that our classification was less effective at delimitat-

ing smaller patches of *G. dewevrei* forest. Sentinel-2 data do have the potential to distinguish between small patches of highly fragmented forest types (e.g., [82]). Our training data are of note here, as some of the field data did come from small patches of *G. dewevrei*, but largely the training data from satellite imagery were skewed towards larger areas of *G. dewevrei* as they are easier to delimitate with a high confidence level. Collecting more field data from small patches of *G. dewevrei* forest may therefore improve the sensitivity of our classifier. Our probability-based classification also showed more uncertainty around the boundaries of *G. dewevrei* forest. This was unexpected since *G. dewevrei* is reported as forming extremely sharp boundaries with mixed forest [44,83].

Efforts to improve sensitivity of the classification could also investigate other remote sensing products; however, a key issue is the availability of data across Central Africa. Hyperspectral sensors capture reflected radiation in narrower spectral bands than multi-spectral sensors (around 10 nm), capturing fine spectral differences across a broad range of wavelengths [84]. Recent advances in hyperspectral sensors such as EnMAP [85] may offer new possibilities for mapping forest types. Very high-resolution images (VHRI) have a spatial resolution of less than 0.5 m [84]. These images have shown promise in identifying individual tree species, enabling the extraction of structural information from individual tree crowns [86,87]. Barbier et al. [52] report that mapping of *G. dewevrei* pockets based on canopy texture descriptors works well for Pleiades, GeoEye, and Worldview images. However, their high costs restrict their use to small-scale studies where budget is available [88].

Recent progress has also been made in distinguishing different forest types using LiDAR and radar data [84,89], which can penetrate the forest canopy, allowing the measurement of structural aspects of the vegetation such as forest height, canopy openness, and density [90–93]. LiDAR data have been used to improve tree species classifications [94–96], and to map vegetation types in a tropical forest-savanna matrix in Gabon, including monodominant Okoumé (*Aucoumea klaineana*) forest [97]. Airborne radar and LiDAR can provide higher resolution data, but these technologies lack global coverage and can be prohibitively expensive. Space-born LiDAR data readily available from the GEDI mission could be investigated to improve classification accuracy, as monodominant *G. dewevrei* forest has structural differences from mixed *terre firme* forest [51]. Sentinel-1 Synthetic-aperture Radar (SAR) data also provide consistent measurements almost unaffected by cloud cover. However, initial inclusion of VV and VH bands from Sentinel-1 images in preliminary classifications for this study showed very low variable importance scores and so were excluded from the final classification.

There is potential for applying this classification to a different study area or scaling up to cover a larger region. Additional considerations, which we were not able to address in the scope of this study, include how well the classifier would distinguish between other tree species that also form monodominant stands, such as *Julbernardia seretii*, especially in the DRC. However, *G. dewevrei* is uniquely dominant in central Africa, reaching higher levels of dominance than any other species [45,50,98], and so may form more uniform canopies than other species, and thus still be distinguished by the Sentinel-2 classifier. Additionally, further investigation into other tree species which form monodominant stands may reveal traits that can be used to separate them from *G. dewevrei*, such as any phenological differences. Our study also focused only on *G. dewevrei* forest within protected areas where human disturbance is minimized. Further studies and data are needed to investigate whether *G. dewevrei* forest can be separated from degraded and disturbed areas and from tree plantations using the methods developed here.

The research presented in this study also indicates exciting possibilities in the field of mapping individual tree species, which is an area of high interest in the current remote sensing research environment, e.g., [35,99]. However, a key feature of the monodomi-

nant stands we were able to map was the uniform texture created by the many adjacent crowns of the same species, which may limit its applicability to species that do not form monodominant stands of this kind.

## 5. Conclusions

Our findings show that Sentinel-2 data can successfully map a tropical forest type within a matrix of spectrally very similar forest types. Monodominant *Gilbertiodendron dewevrei* forest was mapped across the Sangha Trinational using Sentinel-2 data, and a combination of standard spectral bands, vegetation indices, and GLCM textural metrics. The textural metrics proved to be the most important factors for delimitating this forest type, due to the more regular texture created by crowns of *G. dewevrei* being consistently different from the more varied texture created by the high numbers of different species in mixed *terre firme* forest. We have also shown that within the Sangha Trinational, unlike in other areas, *G. dewevrei* forest is not just distributed alongside rivers, but a large amount of *G. dewevrei* forest exists in large upland patches. This presents exciting avenues for future research focusing on a better understanding of the ecology of this species. Our analysis produced a local map of *G. dewevrei* forest, which will be useful for informing conservation management of the Sangha Trinational, and more generally demonstrated the potential of Sentinel-2 data, and in particular that of textural metrics, in mapping different forest types across central Africa, if these exhibit structural and textural differences, such as seen in monodominant and mixed forest types.

**Supplementary Materials:** The following supporting information can be downloaded at: https://www.mdpi.com/article/10.3390/rs17091639/s1; File S1: Shapefiles for maps of *Gilbertiodendron dewevrei* forest across the Sangha Trinational.

**Author Contributions:** Conceptualization, E.H., A.A. and D.J.H.; methodology E.H., A.A. and D.J.H.; investigation: E.H., D.J.H. and J.M.; formal analysis E.H. and A.A.; resources: E.H., D.J.H., D.M. and C.S.; supervision A.A. and D.J.H.; validation A.A. and E.H.; writing—original draft preparation E.H., writing—review and editing A.A., D.J.H., J.M., D.M. and C.S. All authors have read and agreed to the published version of the manuscript.

**Funding:** This research was funded by the Natural Environment Research Council (NERC), through the E4 Doctoral Training Partnership, project reference number 2613926.

**Data Availability Statement:** The Earth Observation datasets that supported the findings of this study are publicly available (for example, Google Earth Engine data catalogue). The maps of *Gilbertiodendron dewevrei* produced here are available from Supplementary Materials. The code is available on Google Earth Engine: https://code.earthengine.google.com/?accept_repo=users/ellenheimpel/Mapping_Monodominant_Forest_2025 (accessed on 5 May 2025). Reference data points are available https://doi.org/10.5281/zenodo.15342714 (accessed on 5 May 2024).

**Acknowledgments:** We thank the Ministère de l'Economie Forestière and the Ministere de le Recherche Scientifique et de l'Innovation Technique of the Government of Congo for their permission to carry out the field research. We are also grateful to the Agence Congolaise de la Faune et des Aires Protégées (ACFAP) for their continued collaboration. Permissions were provided by Institut National de Recherche Forestière, under research authorization number 139, dated 10 August 2023. The Wildlife Conservation Society's Congo Program and the Nouabalé-Ndoki Foundation are acknowledged for their integral partnership in this research. The Royal Botanic Garden Edinburgh is supported by the Scottish Government's Rural and Environment Science and Analytical Services Division. E.H. received a stipend from the NERC DTP E4 to the University of Edinburgh. We also thank Kyle Dexter for useful discussions, and Yunxia Wang for providing advice on mapping using Google Earth Engine.

**Conflicts of Interest:** The authors declare no conflicts of interest.

## Abbreviations

The following abbreviations are used in this manuscript:

| | |
|---|---|
| GLCM | Gray Level Co-occurrence Matrix |
| SVM | Support Vector Machine |
| PCA | Principal Component Analysis |
| SAVG | Sum Average |
| BRDF | Bidirectional Reflectance Distribution Function |
| HAND | Height Above Nearest Drainage |
| VHRI | Very High-Resolution Images |
| SAR | Synthetic-Aperture Radar |
| ESA | European Space Agency |

## Appendix A

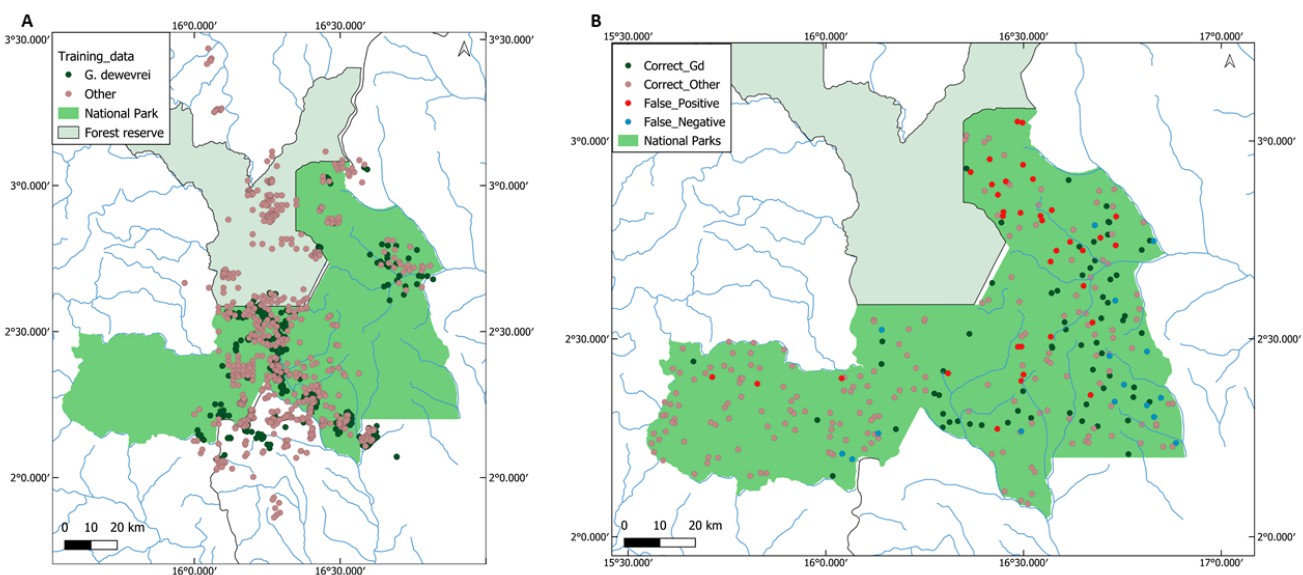

**Figure A1.** Training and validation datasets. (**A**) Dataset of *Gilbertiodendron dewevrei* points and points in other categories of tree cover, used to train the random forest algorithm. Green points are *G. dewevrei* and pink points are other vegetation types. (**B**) Randomly generated validation dataset, plotted according to results from accuracy assessment. Green points are those correctly classified as *G. dewevrei*, pink points are those correctly classified as other, red points are false positives (classified as *G. dewevrei* but validated as other), and blue points are false negatives (classified as other but validated as *G. dewevrei*).

## Appendix B

**Table A1.** Accuracy assessment for mapping of *G. dewevrei* forest across the Sangha Trinational according to methods proposed by Olofsson et al. [69] if all points in our validation dataset of which we were uncertain had the opposite class to what was assigned.

| | | **(A) Error Matrix of Reference Data** | | | | |
|---|---|---|---|---|---|---|
| | | **Reference** | | | | |
| | | *G. dewevrei* | Other | **Total** | **Area** | **Precision** |
| **Map** | *G. dewevrei* | 68 | 18 | 86 | 228,700 | 0.79 |
| | Other | 32 | 182 | 214 | 557,100 | 0.85 |
| | **Total** | 100 | 200 | 300 | 785,800 | |
| | **Recall** | 0.68 | 0.91 | | | |
| | **F1 score** | 0.73 | 0.88 | | | |

**Table A1.** *Cont.*

| | | (B) Error Matrix of Area Proportions | | | | |
|---|---|---|---|---|---|---|
| | | Reference | | | | |
| | | *G. dewevrei* | Other | **Total** | **User accuracy** | **95% CI** |
| **Map** | *G. dewevrei* | 0.23 | 0.061 | 0.291 | 0.79 | 0.09 |
| | Other | 0.106 | 0.603 | 0.709 | 0.85 | 0.05 |
| | **Total** | 0.336 | 0.664 | 1 | | |
| | **Producer accuracy** | 0.68 | 0.91 | **Overall** | **0.83** | **0.04** |
| | **95% CI** | 0.07 | 0.03 | | | |

| | (C) Mapped and Estimated Area | | | |
|---|---|---|---|---|
| | **Mapped area (ha)** | **Estimated area (ha)** | **Lower 95% CI** | **Upper 95% CI** |
| *G. dewevrei* | 228,700 | 264,137 | 230,925 | 297,349 |
| Other | 557,100 | 521,663 | 488,451 | 554,875 |
| **Total** | 785,800 | | | |

## Appendix C

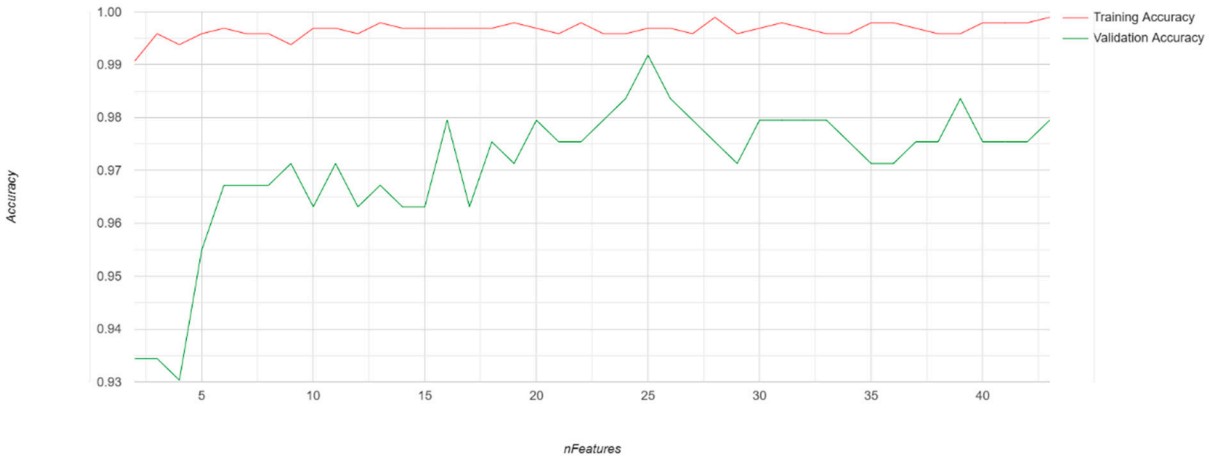

**Figure A2.** Training and validation accuracies from recursive feature elimination (RFE) process. Initial classification was run iteratively, removing the least important variable at each step. Training accuracies and validation accuracies based on a withheld 20% subset of the training data are plotted.

## Appendix D

**Table A2.** Sensitivity analysis for number of trees used in random forest algorithm for classifying monodominant *G. dewevrei* forest. Table reports number of trees used, training accuracy of initial classification, and validation accuracy based on a withheld 20% subset of the training data.

| Number of Trees | Training Accuracy | Validation Accuracy |
|---|---|---|
| 100 | 0.996 | 0.898 |
| 200 | 0.999 | 0.895 |
| 300 | 0.999 | 0.898 |
| 400 | 1 | 0.895 |
| 500 | 1 | 0.902 |
| 600 | 1 | 0.902 |
| 700 | 1 | 0.902 |
| 800 | 1 | 0.906 |
| 900 | 1 | 0.906 |
| 1000 | 1 | 0.906 |

## Appendix E

**Table A3.** Accuracy assessment for mapping of *G. dewevrei* forest across the Sangha Trinational according to methods proposed by Olofsson et al. [69] for the region of the classification influenced by striping in the Sentinel-2 imagery.

| | | **(A) Error Matrix of Reference Data** | | | | |
|---|---|---|---|---|---|---|
| | | | **Reference** | | | |
| | | *G. dewevrei* | Other | **Total** | **Area** | **Precision** |
| **Map** | *G. dewevrei* | 31.00 | 3.00 | 34 | 84,800 | 0.91 |
| | Other | 7.00 | 39.00 | 46 | 163,900 | 0.85 |
| | **Total** | 38.00 | 42.00 | 80 | 248,700 | |
| | **Recall** | 0.82 | 0.93 | | | |
| | **F1 score** | 0.86 | 0.89 | | | |
| | | **(B) Error Matrix Of area Proportions** | | | | |
| | | | **Reference** | | | |
| | | *G. dewevrei* | Other | **Total** | **User accuracy** | **95% CI** |
| **Map** | *G. dewevrei* | 0.311 | 0.03 | 0.341 | 0.91 | 0.1 |
| | Other | 0.1 | 0.559 | 0.659 | 0.85 | 0.1 |
| | **Total** | 0.411 | 0.589 | 1 | | |
| | **Producer accuracy** | 0.76 | 0.95 | **Overall** | **0.87** | **0.08** |
| | **95% CI** | 0.13 | 0.05 | | | |
| | | **(C) Mapped and Estimated Area** | | | | |
| | | **Mapped area (ha)** | **Estimated area (ha)** | **Lower 95% CI** | **Upper 95% CI** | |
| | *G. dewevrei* | 84,800 | 102,259 | 83,201 | 121,317 | |
| | Other | 163,900 | 146,441 | 127,383 | 165,499 | |
| | **Total** | 248,700 | | | | |

**Table A4.** Accuracy assessment for mapping of *G. dewevrei* forest across the Sangha Trinational according to methods proposed by Olofsson et al. [69] for classification excluding the region influenced by striping in the Sentinel-2 imagery.

| | | **(A) Error Matrix of Reference Data** | | | | |
|---|---|---|---|---|---|---|
| | | | **Reference** | | | |
| | | *G. dewevrei* | Other | **Total** | **Area** | **Precision** |
| **Map** | *G. dewevrei* | 34.00 | 13.00 | 47 | 143,900 | 0.72 |
| | Other | 28.00 | 145.00 | 173 | 393,200 | 0.84 |
| | **Total** | 62.00 | 158.00 | 220 | 537,100 | |
| | **Recall** | 0.55 | 0.92 | | | |
| | **F1 score** | 0.62 | 0.88 | | | |
| | | **(B) Error Matrix of Area Proportions** | | | | |
| | | | **Reference** | | | |
| | | *G. dewevrei* | Other | **Total** | **User accuracy** | **95% CI** |
| **Map** | *G. dewevrei* | 0.194 | 0.074 | 0.268 | 0.72 | 0.13 |
| | Other | 0.118 | 0.614 | 0.732 | 0.84 | 0.06 |
| | **Total** | 0.312 | 0.688 | 1 | | |
| | **Producer accuracy** | 0.62 | 0.89 | **Overall** | **0.81** | **0.05** |
| | **95% CI** | 0.09 | 0.05 | | | |
| | | **(C) Mapped and Estimated Area** | | | | |
| | | **Mapped area (ha)** | **Estimated area (ha)** | **Lower 95% CI** | **Upper 95% CI** | |
| | *G. dewevrei* | 143,900 | 167,737 | 139,199 | 196,275 | |
| | Other | 393,200 | 369,363 | 340,825 | 397,901 | |
| | **Total** | 537,100 | | | | |

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
