# Peer review of "Mapping of Monodominant Gilbertiodendron dewevrei Forest Across the Western Congo Basin Using Sentinel-2 Imagery"

_remotesensing, doi:10.3390/rs17091639_

Round 1

Reviewer 1 Report

Comments and Suggestions for Authors

This study presents a valuable application of Sentinel-2 data for mapping Gilbertiodendron monodominant forests in the Congo Basin, addressing an important gap in tropical forest classification. The methodological approach is sound, leveraging spectral, vegetation, and textural indices with a Random Forest classifier. The findings are relevant for conservation and ecological monitoring, particularly in biodiverse regions like the Sangha Trinational area. However, some aspects of the manuscript could be improved for clarity, methodological rigor, and broader applicability.

  1. Lines 39-95: The introduction lacks depth, with only 29 references. Please expand to over 50 references.
  2. Figure 1 is not visually appealing. Please revise it.
  3. How was the independence between the training and test datasets ensured? Were spatial or temporal cross-validation techniques employed to avoid overfitting?
  4. This study focuses on a protected area (Sangha Tri-National). Would the model perform equally well in non-protected or degraded areas, especially where G. dewevrei might coexist with human disturbances?Was multi-temporal Sentinel-2 data considered to account for phenological variations (e.g., leaf flushing/flowering periods) that might improve discrimination?
  1. How does this method compare with other remote sensing approaches (e.g., LiDAR, hyperspectral) in mapping monodominant forests? A brief cost-effectiveness discussion would enhance the practical relevance of the paper.

Author Response

This study presents a valuable application of Sentinel-2 data for mapping Gilbertiodendron monodominant forests in the Congo Basin, addressing an important gap in tropical forest classification. The methodological approach is sound, leveraging spectral, vegetation, and textural indices with a Random Forest classifier. The findings are relevant for conservation and ecological monitoring, particularly in biodiverse regions like the Sangha Trinational area. However, some aspects of the manuscript could be improved for clarity, methodological rigor, and broader applicability.

Lines 39-95: The introduction lacks depth, with only 29 references. Please expand to over 50 references.

Thank you, we have now revised the introduction, adding in more detail and references.

Figure 1 is not visually appealing. Please revise it.

We have made amendments to Figure 1 to address this comment. It is not clear which precise aspects of the figure the reviewer would like us to change, but we have removed the fill colour and added in images of our final classification maps to add visual interest.

How was the independence between the training and test datasets ensured? Were spatial or temporal cross-validation techniques employed to avoid overfitting?

Our test data was completely independent from the training dataset. We created an additional novel dataset to test our classifier. This was done by performing disproportional stratified random sampling, randomly sampling 200 points in “other forest” and 100 points in “G. dewevrei forest” from our classified map. These points were then independently classified according to high-resolution satellite imagery, and compared to the map classification to calculate accuracy and to estimate area figures. Our description of this process has been amended slightly (line 264), to add clarification.  A map of the two separate datasets is presented in Appendix A (Figure A1, lines 620 - 628).

This study focuses on a protected area (Sangha Tri-National). Would the model perform equally well in non-protected or degraded areas, especially where G. dewevrei might coexist with human disturbances? Was multi-temporal Sentinel-2 data considered to account for phenological variations (e.g., leaf flushing/flowering periods) that might improve discrimination?

Thank you for the suggestion of using multi-temporal Sentinel-2 data to examine phenological variations, which we did explore in our initial planning for this study. We created Sentinel-2 composites across a region with a known boundary between G. dewevrei forest and mixed terre firme forest each month for the year 2020. We did not observe any distinct phenological differences of the G. dewevrei forest across the year. G. dewevrei is evergreen, so retains its leaves year-round. We therefore decided to instead create a multi-year Sentinel-2 composite (2020 - 2021) to run our classification on, in order to reduce the effects of cloud cover, and artifacts caused by atmospheric correction models in cloudy regions. We have added mention of this into the manuscript (lines 183-184).

Thank you for raising the question of whether the model would perform equally well in non-protected or degraded areas. We have added this question into the discussion section (lines 555-558), but unfortunately testing this is out of the scope of our study, as we do not have sufficient ground referenced data of G. dewevrei in areas of human disturbance.

How does this method compare with other remote sensing approaches (e.g., LiDAR, hyperspectral) in mapping monodominant forests? A brief cost-effectiveness discussion would enhance the practical relevance of the paper.

This is a good suggestion, and we have now added a section on this into the discussion (lines 520-545). In summary other remote sensing approaches could offer further possibilities for mapping monodominant forests which should be investigated. VHRI and airborne LiDAR have shown promise for improving tree species classifications from space, however they are very expensive and lack global coverage. Space-born LiDAR data from GEDI have the potential to add to the classification accuracies of mapping monodominant forest types. Sentinel-1 C-band synthetic aperture radar (VV and VH polarisation) proved uninformative in an initial classification conducted for this study, and was therefore excluded.

Reviewer 2 Report

Comments and Suggestions for Authors

This study proposed a new method for mapping Gilbertiodendron dewevrei monodominant forests using Sentinel-2 data, which is of great significance for the protection of tropical ecosystems and remote sensing applications. However, there are several questions as follows:

Too many repeated keywords (e.g. Congo Basin; Tropical Forests), non-core terms occupy keywords (e.g. Cloud Computing. This study mainly uses Google Earth Engine as a data processing tool, but does not involve innovative analysis of cloud computing technology. It is recommended to delete it.)

Line 40, remote sensing should be cited for the progress and challenges in forest type classification and mapping.

Lines 46 to 48, "Changes in species assemblages and ecosystem function due to deforestation may go undetected, hampering targeted interventions for their conservation." It feels unrelated to the article.

Section 2.3, page 6

The recursive feature elimination (RFE) process lacks clarity. Specify the threshold or criteria for retaining 16 input variables (Table 1). This is critical for reproducibility

Section 2.4, page 7

The selection of 500 decision trees lacks justification. A sensitivity analysis (e.g., testing the number of trees from 100 to 1,000) should be provided to demonstrate parameter optimization. In addition, the manuscript should explain why random forests were chosen over other classifiers (e.g., SVM or deep learning). A brief comparison with other methods would strengthen the methodological rationale.

Section 2.2.2, part

“ESA WorldCover 10m 2020 product”, does not state whether the accuracy of this product in tropical rainforests has been validated. Relevant validation studies should be cited.

Page 11, the accuracy assessment of the striped area (Figure 6) only mentions “consistent with the whole”, but no quantitative comparison data is provided. The independent validation results of the striped and non-striped areas need to be supplemented. Page 14, although the striped problem was mitigated by feature screening, it is not stated whether the impact on classification was completely eliminated. The potential risk of residual effects on regional extrapolation needs to be discussed.

Page 14, Line 431, mentions that "the taproot of G. dewevrei can reach 2m", but no supporting literature is cited. Additional citations of relevant research are needed.

Author Response

This study proposed a new method for mapping Gilbertiodendron dewevrei monodominant forests using Sentinel-2 data, which is of great significance for the protection of tropical ecosystems and remote sensing applications. However, there are several questions as follows:

Too many repeated keywords (e.g. Congo Basin; Tropical Forests), non-core terms occupy keywords (e.g. Cloud Computing. This study mainly uses Google Earth Engine as a data processing tool, but does not involve innovative analysis of cloud computing technology. It is recommended to delete it.)

The wording has been amended as advised.

Line 40, remote sensing should be cited for the progress and challenges in forest type classification and mapping.

This is a good point that this sentence previously lacked support from the literature. We have now added in references to reviews on the progresses and challenges in forest type classification and mapping using remotely sensed imagery (Line 41).

Lines 46 to 48, "Changes in species assemblages and ecosystem function due to deforestation may go undetected, hampering targeted interventions for their conservation." It feels unrelated to the article.

We have reworded this section to improve clarity (Lines 43-50). We respectfully disagree with the notion that the sentence is unrelated to the study. It provides important context and a justification for this study: an ability to map forest types within the Congo Basin Forest block is necessary for targeted conservation assessments and efforts.

Section 2.3, page 6

The recursive feature elimination (RFE) process lacks clarity. Specify the threshold or criteria for retaining 16 input variables (Table 1). This is critical for reproducibility

Thank you, this is a good point. We have added details on the RFE process, including the threshold used for retaining input variables (Line 219-224), and a figure (Figure A2, Appendix C, Lines 654-660) illustrating how validation accuracies changed with different numbers of input variables.

Section 2.4, page 7

The selection of 500 decision trees lacks justification. A sensitivity analysis (e.g., testing the number of trees from 100 to 1,000) should be provided to demonstrate parameter optimization. In addition, the manuscript should explain why random forests were chosen over other classifiers (e.g., SVM or deep learning). A brief comparison with other methods would strengthen the methodological rationale.

This is a good point. We have now added details on a sensitivity analysis, which justifies the choice of 500 decision trees to the main text (lines 246-251) and the appendices (Appendix D, lines 662-666). We have also added a section explaining the choice of Random Forests over other classifiers (lines 237-245). In summary RF was chosen over deep learning due to the computational intensity and methodological complexity of deep learning, and over SVM as it allows the study of variable importance, it does not come with the complexity of choosing a suitable kernel, and was reported as achieving a higher accuracy for studies of LULC.

Section 2.2.2, part

“ESA WorldCover 10m 2020 product”, does not state whether the accuracy of this product in tropical rainforests has been validated. Relevant validation studies should be cited.

This is a good point. The ESA tree cover map achieved an overall accuracy of 74.4 (73.6 ±0.2 95% CI in Africa) with a reasonably good accuracy for tree cover (User’s accuracy of 80.1 ±0.1 and Producer’s accuracy of 89.9 ±0.1). Thus, there was a slight overestimation of tree cover. The study area is mainly tree cover, so we expect this to have minimal impacts on the map presented here. However, if the classification is expanded to other areas which include areas of non-tree cover misclassified by the ESA tree cover map, these could potentially include those displaying similar characteristics to G. dewevrei forest, for example homogenous texture. We have added a reference to this independent validation study into the main text (lines 189-192).

Page 11, the accuracy assessment of the striped area (Figure 6) only mentions “consistent with the whole”, but no quantitative comparison data is provided. The independent validation results of the striped and non-striped areas need to be supplemented. Page 14, although the striped problem was mitigated by feature screening, it is not stated whether the impact on classification was completely eliminated. The potential risk of residual effects on regional extrapolation needs to be discussed.

Thank you for pointing this out. We have now added the independent validation results of the striped and non-striped areas into the appendices (Appendix E, Tables A3 and A4 lines 667 – 693). The overall accuracies of both of these regions (stiped and non-striped) have overlapping confidence intervals with the overall accuracy reported in the manuscript. The striped region actually shows a higher overall accuracy, and notably a higher user’s accuracy and producers’ accuracy for G. dewevrei forest. By conducting feature screening, and including training and test data within the striped region, we have minimised the potential residual influence of the striping in Sentinel-2 imagery on the classification presented here.

Risk of residual effects on regional extrapolation are explored in the discussion, highlighting that when the classification is extended to areas where less training data is available, residual effects become visually apparent (lines 495-500). We advise caution in extending this classification to areas where insufficient training data is available.

Page 14, Line 431, mentions that "the taproot of G. dewevrei can reach 2m", but no supporting literature is cited. Additional citations of relevant research are needed.

Thanks for pointing this out. Relevant references have now been added in (line 475).  
